# Analysis of the Whole-Exome Sequencing of Tumor and Circulating Tumor DNA in Metastatic Melanoma

**DOI:** 10.3390/cancers11121905

**Published:** 2019-11-29

**Authors:** Russell J. Diefenbach, Jenny H. Lee, Dario Strbenac, Jean Y. H. Yang, Alexander M. Menzies, Matteo S. Carlino, Georgina V. Long, Andrew J. Spillane, Jonathan R. Stretch, Robyn P. M. Saw, John F. Thompson, Sydney Ch’ng, Richard A. Scolyer, Richard F. Kefford, Helen Rizos

**Affiliations:** 1Department of Biomedical Sciences, Faculty of Medicine and Health Sciences, Macquarie University, Sydney, NSW 2109, Australia; russell.diefenbach@mq.edu.au (R.J.D.); jenny.lee@mq.edu.au (J.H.L.); 2Melanoma Institute Australia, The University of Sydney, Sydney, NSW 2065, Australia; alexander.menzies@sydney.edu.au (A.M.M.); matteo.carlino@sydney.edu.au (M.S.C.); georgina.long@sydney.edu.au (G.V.L.); andrew.spillane@sydney.edu.au (A.J.S.); jonathan.stretch@melanoma.org.au (J.R.S.); robyn.saw@melanoma.org.au (R.P.M.S.); john.thompson@melanoma.org.au (J.F.T.); sydney.chng@sydney.edu.au (S.C.); richard.scolyer@health.nsw.gov.au (R.A.S.); richard.kefford@mq.edu.au (R.F.K.); 3School of Mathematics and Statistics, The University of Sydney, Sydney, NSW 2006, Australia; dario.strbenac@sydney.edu.au (D.S.); jean.yang@sydney.edu.au (J.Y.H.Y.); 4Charles Perkins Centre, The University of Sydney, Sydney, NSW 2006, Australia; 5Sydney Medical School, The University of Sydney, Sydney, NSW 2006, Australia; 6Department of Medical Oncology, Northern Sydney Cancer Centre, Royal North Shore Hospital, Sydney, NSW 2065, Australia; 7Crown Princess Mary Cancer Centre, Westmead and Blacktown Hospitals, Sydney, NSW 2145, Australia; 8Department of Melanoma and Surgical Oncology, Royal Prince Alfred Hospital, Sydney, NSW 2050, Australia; 9Department of Tissue Pathology and Diagnostic Oncology, Royal Prince Alfred Hospital, Sydney, NSW 2050, Australia; 10Department of Clinical Medicine, Faculty of Medicine and Health Sciences, Macquarie University, Sydney, NSW 2109, Australia

**Keywords:** whole exome sequencing, melanoma, circulating tumor DNA

## Abstract

The use of circulating tumor DNA (ctDNA) to monitor cancer progression and response to therapy has significant potential but there is only limited data on whether this technique can detect the presence of low frequency subclones that may ultimately confer therapy resistance. In this study, we sought to evaluate whether whole-exome sequencing (WES) of ctDNA could accurately profile the mutation landscape of metastatic melanoma. We used WES to identify variants in matched, tumor-derived genomic DNA (gDNA) and plasma-derived ctDNA isolated from a cohort of 10 metastatic cutaneous melanoma patients. WES parameters such as sequencing coverage and total sequencing reads were comparable between gDNA and ctDNA. The mutant allele frequency of common single nucleotide variants was lower in ctDNA, reflecting the lower read depth and minor fraction of ctDNA within the total circulating free DNA pool. There was also variable concordance between gDNA and ctDNA based on the total number and identity of detected variants and this was independent of the tumor biopsy site. Nevertheless, established melanoma driver mutations and several other melanoma-associated mutations were concordant between matched gDNA and ctDNA. This study highlights that WES of ctDNA could capture clinically relevant mutations present in melanoma metastases and that enhanced sequencing sensitivity will be required to identify low frequency mutations.

## 1. Introduction

Analysis of circulating tumor DNA (ctDNA) can be used to identify therapeutically actionable mutations, monitor disease progression, detect residual disease and track the genomic evolution of treatment resistance [1,2,3,4]. The genomic profiling of ctDNA using whole genome, whole exome or targeted sequencing is feasible, but technically challenging due to the low quantities and fragmented nature of ctDNA [5,6,7,8]. Nevertheless, several reports have examined the value of sequencing patient-matched tumor gDNA and ctDNA and the results showed variable concordance in the mutation profiles of gDNA and ctDNA derived from patients with neuroblastoma, breast cancer, lung cancer or gastrointestinal malignancies [9,10,11,12]. The degree of concordance between circulating and tumor mutations appears to reflect tumor heterogeneity, which can be influenced by prior therapy, tumor stage and tumor mutation burden [12].

The NGS profiling of melanoma ctDNA and matched tumor tissue, which has been limited, is of particular interest, as tumor and ctDNA profiling has provided significant predictive and prognostic value. For instance, high tumor mutation burden predicts response, progression-free and overall survival of melanoma patients to immune checkpoint inhibitors. Longitudinal ctDNA assessment predicts progression-free survival and overall survival in stage IV melanoma patients treated with BRAF and MEK inhibitors or immunotherapy [13,14,15] and the melanoma-specific survival of patients with high-risk stage III resected melanoma [16]. ctDNA can also monitor the appearance of treatment-resistant melanoma subclones [14] and differentiate true progression from pseudoprogression in melanoma patients treated with immunotherapy [17]. In the majority of these studies, melanoma ctDNA and tumor tissue were evaluated separately and ctDNA analysis was limited to single mutation detection.

In this study, we examined the whole-exome sequencing (WES) mutation profiles of patient-matched melanoma tissue and matched ctDNA in a cohort of 10 patients to determine whether sequencing of these biopsies could provide complimentary, clinically actionable mutation data. We now confirm that the WES of ctDNA provides valuable mutation data, and is capable of identifying hotspot and other clinically relevant mutations, irrespective of prior therapy, timing of tissue and liquid biopsy site. However, significant enhancement of WES sensitivity is required for ctDNA to provide an accurate and complete profile of tumor subclones.

## 2. Results

### 2.1. Patient Cohort

Ten patients with metastatic melanoma with established driver BRAF or NRAS mutations were included in this study; 5/10 patients were aged 65 years or older, 8/10 were male and 6/10 had American Joint Committee on Cancer (AJCC) stage M1c disease [18] (Table 1). Time between tissue and liquid biopsy was less than 6 months in all patients (median 2.75 months, range 1–6 months), and three patients each had tissue and a liquid biopsy collected concurrently. Samples were collected prior to systemic treatment, i.e., pre-treatment (PRE) or at time of the progression of treatment (PROG). Systemic treatments included the immune checkpoint inhibitors ipilimumab, nivolumab and pembrolizumab and the targeted therapies, dabrafenib and trametinib.

### 2.2. WES and Droplet Digital PCR

The amount of tumor tissue gDNA extracted for this study was in the µg range and significantly higher than the ng amounts of extracted plasma ctDNA (Appendix A). Nevertheless, sequencing libraries were successfully generated for ctDNA and gDNA for each patient, and WES output based on total reads and coverage was not significantly different when comparing gDNA and ctDNA (Figure 1A,B). However, the mean read depth was found to be significantly less for ctDNA, presumably due to the lower amounts of ctDNA template available for sequencing (Figure 1C). This was not always the case, however—an exception being higher ctDNA compared to gDNA mean read depth for patients 4 and 9 (Figure 1C). The coverage distributions of targeted regions by WES were also similar for gDNA and ctDNA (Appendix A).

We also examined the relationship between ctDNA copy number and total circulating free DNA (cfDNA) quantities in the 10 melanoma patients, by measuring the amount of ctDNA using highly sensitive droplet digital (dd) PCR. We found that ctDNA copy number significantly correlated with total extracted cfDNA (Appendix A).

### 2.3. Single Nucleotide Variant Analysis

A list of single nucleotide variants (SNVs) was generated from matched gDNA and ctDNA WES. Non-synonymous SNVs with mutant allele frequency (MAF) ≥10% were included (summarized in Appendix A). There was no significant difference in the total number of SNVs in the patient-matched tumor gDNA and ctDNA and no difference in the derived tumor mutation burden determined from WES of either gDNA or ctDNA (Figure 2).

The degree of overlap or concordance of SNVs for gDNA and ctDNA from matched patient samples ranged from 22.7% to 77.6% (Figure 3). Similar percentages of SNVs were unique to either gDNA (range 13–66.5%) or ctDNA (range 7.7–53.2%) (Figure 3). Overall, there was no detectable difference in the degree of overlap of SNVs in matched gDNA and ctDNA from individual patients regardless of treatment type (naïve, combination dabrafenib and trametinib (combi-DT) or immune checkpoint therapy), response to treatment, timing of biopsies or location of the biopsied tumor (Table 1 and Appendix A). Moreover, the three patients with concurrent tissue and liquid biopsies displayed a wide range of SNV overlap (24.5% in patient 8, 33.8% in patient 6 and 77.6% in patient 1).

### 2.4. Mutant Allele Frequency (MAF) of Single Nucleotide Variants

To confirm that ctDNA sequencing provided accurate allele frequency data, we compared the MAFs for the shared SNVs in patient-matched gDNA and ctDNA. These showed significant correlations in all patients, with the highest level of correlation between gDNA and ctDNA observed in patients 6 and 9 (Figure 4 and Appendix A). Interestingly, both patients had gDNA derived from core liver biopsies (Table 1), and the high concordance between circulating and liver melanoma SNVs may reflect the fact the liver melanoma metastases tend to be larger than melanoma metastases in other sites, including brain, soft tissue and lung [20].

### 2.5. Melanoma Driver Mutations

We then focused on the *BRAF* and *NRAS* melanoma driver gene mutations (Table 1). WES of gDNA was able to identify the driver mutations in all patients (MAF range 25–83%), whereas WES of ctDNA only detected the driver mutation in six of ten patients (patients 1, 3, 5, 6, 7 and 9) when applying a MAF cutoff of at least 10% (with a call quality of at least 20 and read depth of at least 10 as described in Materials and Methods) (Appendix A). However, the ctDNA driver mutations were detected by manual curation in the remaining four patients (patients 2, 4, 8 and 10; MAF 7–12%), and were well below the gDNA MAF (Appendix A). Comparison of the driver MAF determined by WES of gDNA versus ctDNA across all 10 patients showed no significant correlation (Figure 5A). All driver MAFs in ctDNA were independently validated; nine using ddPCR analysis for either *BRAF* or *NRAS* mutations and one using highly sensitive targeted sequencing analysis (Appendix A). There was significant correlation between MAF based on WES and ddPCR/targeted NGS sequencing of ctDNA (Figure 5B). However, there was less correlation (though still significant) when the driver MAF based on WES of gDNA and ddPCR analysis of ctDNA was compared (Figure 5C). This highlights that melanoma driver MAF captured in ctDNA is generally lower than the driver MAF from gDNA, consistent with our observation that MAF of common SNVs was generally lower in ctDNA WES compared to patient-matched gDNA WES data (Figure 4 and Appendix A).

### 2.6. Other Highlighted Mutations

In addition to mutations in the *BRAF* or *NRAS* genes, we examined other melanoma-associated genes (gene list shown in Appendix A [21,22,23,24,25]) or melanoma-associated mutations (based on cbioportal [26,27]) in the WES dataset (Appendix A). These genes or mutations were detected initially in either gDNA, ctDNA or both. SNVs unique to either gDNA or ctDNA were subsequently found by manual curation of WES Bam files to occur in both sources of DNA (Appendix A). Only one mutation, MASP2 R356W, was found to be unique to ctDNA in patient 6 (Appendix A). Interestingly, patient 6, the only treatment naïve patient, had the highest number of melanoma-associated mutations (Appendix A), although this patient did not have the highest number of total SNVs (Figure 3). The CDK4 R24C mutation in the BRAF^V600E^ mutant patient 3 was the only additional melanoma-associated mutation predicted to be a driver mutation (Appendix A). Rare germline mutations in CDK4 at position 24 predispose to melanoma susceptibility [28]. We identified an NRAS Q22K mutation in patient 1 (Appendix A). Although this is an uncommon NRAS variant, it has been reported in a small number of tumors, including melanoma [23], and potently induces MAPK signaling [29]. It is worth noting that although this tumor was progressing on the PD1 inhibitor pembrolizumab (Table 1), this patient had progressed on prior BRAF/MEK inhibitor combination therapy, presumably due to the activating NRAS Q22K mutation. Inactivation mutations in ARID2, which encodes a component of the SWI/SNF chromatin remodeling complex, are observed in melanoma [23], and the nonsense ARID Q1165* mutation was enriched in the ctDNA of patient 8 (Appendix A). The MAP3K5 R256C mutation identified in ctDNA and melanoma gDNA from patient 10 has also been identified in melanoma, and shown to inhibit the pro-death activity of this kinase [30].

## 3. Discussion

In this study we compared the WES of matched gDNA and ctDNA from 10 patients with metastatic melanoma in both treatment naïve patients and patients on systemic molecular or immune therapies. We now report that ctDNA sequencing provides an accurate, albeit incomplete snapshot of the mutation profile and mutation burden of the patient-matched metastatic tumor. The SNVs found in the patient-matched melanoma gDNA and ctDNA overlapped by 22.7% to 77.6%, which is comparable to gDNA and ctDNA mutation concordance data reported for other cancers [9,10,12,31]. A major caveat of this work is the lack of patient-matched germline analyses, and although we applied a stringent bioinformatic filtering approach to enrich for somatic variants, the number of somatic variants and the concordance data may have been overestimated. Nevertheless, the ctDNA mutation profiling provided a valuable mutation profile irrespective of the site of the patient-matched tumor and this was particularly important for brain metastases, given that levels of ctDNA derived from brain metastases are extremely low, possibly as a result of the blood–brain barrier [15,32]. It is likely that ctDNA in these melanoma patients is derived from extracranial tumors, and ctDNA remains informative for brain melanoma because melanoma brain tumors have similar mutational and treatment response profiles to concurrent extracranial tumors [33,34,35,36,37]. It is also worth noting that MAFs of common SNVs were most concordant between ctDNA and liver melanoma metastases.

One recent study found substantial discordance between NGS of matched gDNA and ctDNA in several tumor types (including one melanoma sample) and reported that ctDNA analysis may improve detection of hotspot and actionable mutations compared to gDNA alone [12]. Tissue and circulating DNA mutation concordance was reported to be higher in patients who had received systemic therapy and in cancers with high TMB [12]. In our study, 9/10 patients received some sort of systemic treatment, and although melanoma has a very high TMB [38], the concordance between gDNA and ctDNA mutations was variable. Furthermore, although our cohort consisted of gDNA and ctDNA samples collected at different time points, the timing of sample collection was not associated with mutation concordance in our report and in another similar study [12].

The majority of SNVs detected in our study, including the *BRAF* and *NRAS* driver mutations, showed lower MAFs in ctDNA compared to gDNA, and this was presumably due to the combination of lower read depth for ctDNA and the fact that ctDNA makes up only a fraction of total cfDNA [39]. For a few variants, however, including NRAS Q22K and CDK4 R24C, and the MAFs in ctDNA, were higher or equivalent to the MAFs in tumor tissue, suggesting the subclonal expression of these mutations in the lesion sequenced. We also applied a MAF threshold of at least 10% to define variants in the gDNA and ctDNA sequencing data. In some samples, a lower MAF threshold was required to detect the *BRAF* and *NRAS* driver mutations and allele frequency thresholds need to be optimized for sensitivity and specificity with larger cohort studies. Although ddPCR analysis of ctDNA was more sensitive that WES ctDNA sequencing, the additional information gained by unbiased sequencing, including tumor mutation burden, which is associated with clinical benefit for immunotherapy [31] and the potential to define resistant subclones, is an important advantage. For instance, we identified an activating and putative resistance *NRAS* mutation (Q22K) in both gDNA and ctDNA in a BRAF-mutant melanoma after progression on combined BRAF and MEK inhibitors [40]. There is also significant scope to improve the sensitivity of ctDNA sequencing with multi-gene panels, molecular barcoding and size selection enrichment.

In conclusion, WES analysis of total SNVs in ctDNA provides a valuable snapshot of the mutational landscape and burden of melanoma metastases, although additional improvements in sensitivity are required, especially for the analysis of smaller-sized descendent tumor subclones. This may be clinically important as pre-existing variants capable of promoting treatment resistance may cause a fitness deficit and produce only minor descendent clones in the absence of selective pressure. This appears to be the case, for instance, for BRAF V600E gene amplification, a common mediator of resistance to BRAF and MEK inhibitors [41], which also promotes proliferative arrest in the absence of MAPK inhibitors. Thus, the use of ctDNA to identify the tumor metagenome (i.e., the aggregate of co-existent tumor subclones) will require sequencing depth that allows the detection of 0.1% MAF afforded by targeted gene panels [42,43]. At this MAF threshold, ctDNA is detectable in 75% of metastatic melanoma patients [13], and improving the sensitivity of ctDNA sequencing promises significant benefits, especially as a predictive biomarker in an era with many effective therapies.

## 4. Materials and Methods

### 4.1. Human Melanoma Samples

The fresh-frozen tissue and blood samples used in the current study were obtained from the Melanoma Institute Australia biospecimen bank with written informed patient consent and institutional review board approval (Sydney Local Health District Human Research Ethics Committee, Protocol Numbers X15–0454 and HREC/11/RPAH/444). All samples were pathologically assessed prior to inclusion into the study, as previously described [44].

Blood (10 mL) was collected in EDTA tubes (Becton Dickinson, Franklin Lakes, NJ, USA) and processed immediately. Tubes were spun at 800× *g* for 15 min at room temperature. Plasma was then removed into new 15 mL tubes without disturbing the buffy coat and respun at 1600× *g* for 10 min at room temperature to remove cellular debris. Plasma was stored in 1 mL aliquots at −80°C.

### 4.2. DNA Extractions

Tumor-derived gDNA for WES was extracted from fresh-frozen tissue using the DNeasy Blood and Tissue Kit (Qiagen, Hilden, Germany) according to the manufacturer’s instructions. Plasma ctDNA from melanoma patients was purified using the QIAamp circulating nucleic acid kit (Qiagen, Hilden, Germany) according to the manufacturer’s instructions. ctDNA was purified from 3–5 mL of plasma, and the final elution volume was 30–40 µL.

### 4.3. Analysis of Purified ctDNA from Plasma

The copy number of ctDNA extracted from plasma was analyzed as previously described [15]. The QX200 ddPCR system was used to detect mutant *NRAS* or *BRAF* as previously described (Bio-Rad, Hercules, CA, USA) [15]. The copy number was determined with Quantasoft software version 1.7.4 (Bio-Rad, Hercules, CA, USA) using a manual threshold setting. One patient ctDNA sample was analyzed by Thermofisher Scientific Australia using their lung oncomine cfDNA assay according to the manufacturer’s instructions.

### 4.4. Whole Exome Sequencing

Exome library preparation and sequencing workflow utilized the SureSelect V5-post capture kit (Agilent, Santa Clara, CA, USA) and Illumina (San Diego, CA, USA) HiSeq 4000 platform and was performed by the Macrogen corporation (Seoul, South Korea). Total DNA was quantified using a Quant-iT Picogreen dsDNA assay kit (Life Technologies, Carlsbad, CA, USA). The size of sequencing library preparations was ascertained on a 2100 Bioanalyzer (Agilent, Santa Clara, CA, USA) using a DNA 1000 chip. The quantity of individual sequencing libraries was determined using qPCR quantification (Illumina, San Diego, CA, USA) according to the manufacturer’s instructions. Sequencing reads were mapped using Burrows-Wheeler Aligner (BWA) [45] against the human reference genome hg19 (https://genome.ucsc.edu). SNPs were detected using SAMTools [46], variant database dbSNP [47] and 1000 genomes project [48]. Whole exome sequence data have been deposited in the Sequence Read Archive database, Accession PRJNA592419.

### 4.5. Filtered Single Nucleotide Variants

Filtered SNV annotation and interpretation analyses were generated through the use of Ingenuity Variant Analysis software (https://www.qiagenbioinformatics.com/product-login/) from Qiagen (Hilden, Germany).

Since no matched normal tissue or germline DNA was available, we applied a stringent mutation filtering pipeline based on a recently validated somatic mutation filtering strategy [49]. Included SNVs were those with a call quality of at least 20, a read depth of at least 10 and allele-frequency of at least 10. Included SNVs were exonic or the first 2 bases of introns flanking exons, and were classified as disease-associated according to the human gene mutation database (HGMD) or ClinVar or established gain of function or frameshift, indels, or stop-codon or missense mutation unless predicted to be innocuous by SIFT or Polyphen. Included SNVs were also those outside the top 5.0% most exonically variable 100 base windows in healthy public genomes (1000 genomes project) and outside the top 1% most exonically variable genes in the 1000 genomes project [48]). Excluded SNVs that were likely to be polymorphisms were those observed with an allele frequency greater than or equal to 1.0% of the genomes in the 1000 genomes project [48], NHLBI ESP exomes [50], the ExAC dataset [51] or gnomAD [51]. Excluded SNVs included those present in at least 20% of the gDNA or ctDNA patient samples (excluding driver mutations) that are experimentally observed to be associated with a possibly benign phenotype. Further manual filtering consisted of including only exonic SNVs that were non-synonymous mutations and excluding any SNVs still present in the 1000 genomes project.

Melanoma-associated genes were identified using several sources [21,22,23,24,25]. Melanoma-associated mutations were confirmed using cbioportal [26,27] (http://www.cbioportal.org). The tumor mutational burden (SNVs/Mbase) was calculated using a human exome size (Mbase) based on the number of on-target genotypes (with more than 10× sequencing depth) determined for each patient sample subjected to WES and the total number of filtered SNVs.

### 4.6. Statistical Analysis

Venn diagrams were generated using the interactive online tool Venny 2.1 http://bioinfogp.cnb.csic.es/tools/venny/index.html. Pearson correlation coefficients were determined using Prism version 8.1.1. A paired, nonparametric Wilcoxon test was also performed using Prism.

## 5. Conclusions

In this study, we showed that sequencing of ctDNA to capture the mutational profiles and mutation burdens of melanoma metastases is a viable option. The sequencing read depth of ctDNA was consistently lower than gDNA, and thus mutation detection sensitivity is an important limitation, particularly in patients with low tumor burden or intracranial disease. Although WES was used in the current study, this approach is likely to be superseded by targeted deep sequencing due its greater sensitivity accomplished through greater read depth. This is certainly feasible for melanoma, where a targeted gene panel can be designed to cover the majority of known somatic mutations.

## Figures and Tables

**Figure 1 cancers-11-01905-f001:**
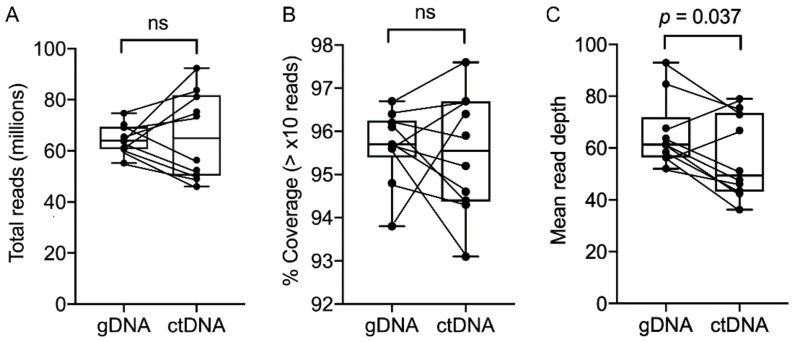
Summary of whole-exome sequencing (WES) analysis. Comparison of (**A**) total reads, (**B**) percentage of coverage (>10× reads) and (**C**) mean read depth for matched (connected by lines) genomic DNA (gDNA) and circulating tumor DNA (ctDNA) from advanced melanoma patients. Plots show median with interquartile ranges and data were compared using the paired, nonparametric Wilcoxon test. Abbreviations: ns, not significant.

**Figure 2 cancers-11-01905-f002:**
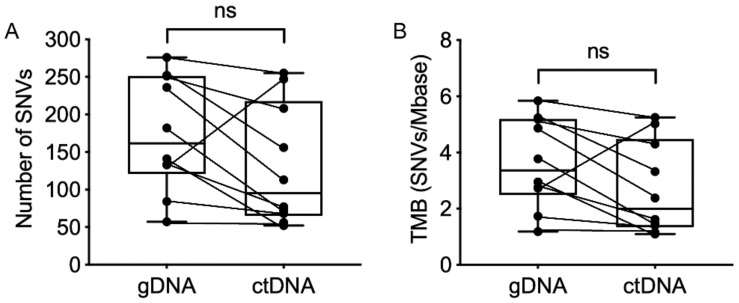
Comparison of the mutational load for melanoma patient matched (connected by lines) genomic DNA (gDNA) and circulating tumor DNA (ctDNA) analyzed by WES. (**A**) Total number of filtered single nucleotide variants (SNVs). (**B**) Tumor mutational burden (TMB). The filtered list of SNVs was produced from the WES list of total SNVs using ingenuity variant analysis, as described in Materials and Methods. The TMB was calculated as described in Materials and Methods. Plots show medians with interquartile ranges and data were compared using the paired, nonparametric Wilcoxon test. Abbreviations: ns, not significant.

**Figure 3 cancers-11-01905-f003:**
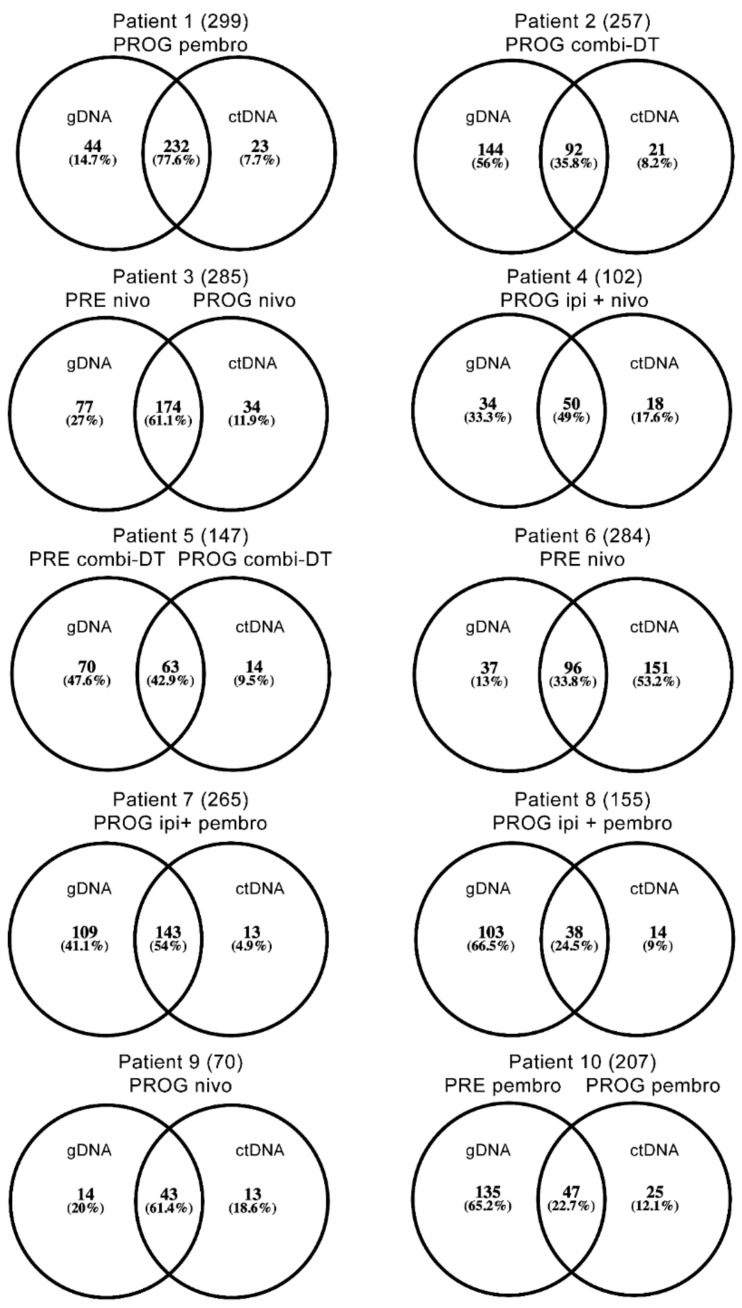
Degree of SNV overlap for patient matched genomic DNA (gDNA) and circulating tumor DNA (ctDNA) as identified by WES. The filtered list of SNVs was produced from the WES list of total SNVs using an ingenuity variant analysis as described in Materials and Methods. The numbers in parentheses represent total combined SNVs. Abbreviations: PRE, prior to systemic therapy; PROG, at time of progression of treatment; nivo, nivolumab (anti-PD1); pembro, pembrolizumab (anti-PD1); ipi, ipilimumab (anti-CTLA4); combi-DT, combination dabrafenib (BRAF inhibitor) and trametinib (MEK inhibitor).

**Figure 4 cancers-11-01905-f004:**
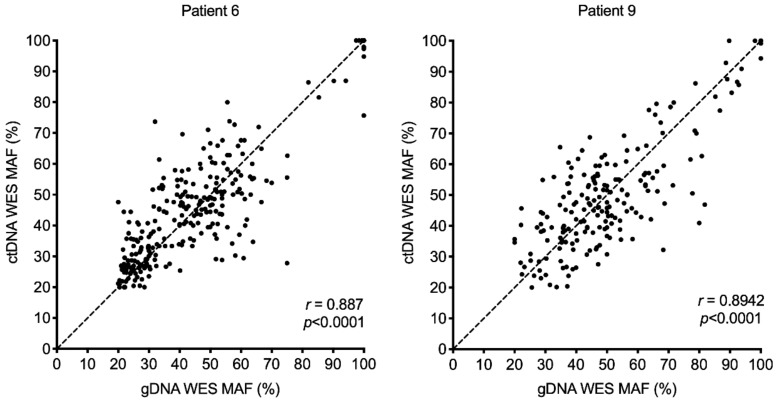
Degree of Pearson correlation between the mutant allele frequency (MAF) of SNVs common to patient matched genomic DNA (gDNA) and circulating tumor DNA (ctDNA), as identified by WES. Dotted line indicates *y* = *x*.

**Figure 5 cancers-11-01905-f005:**
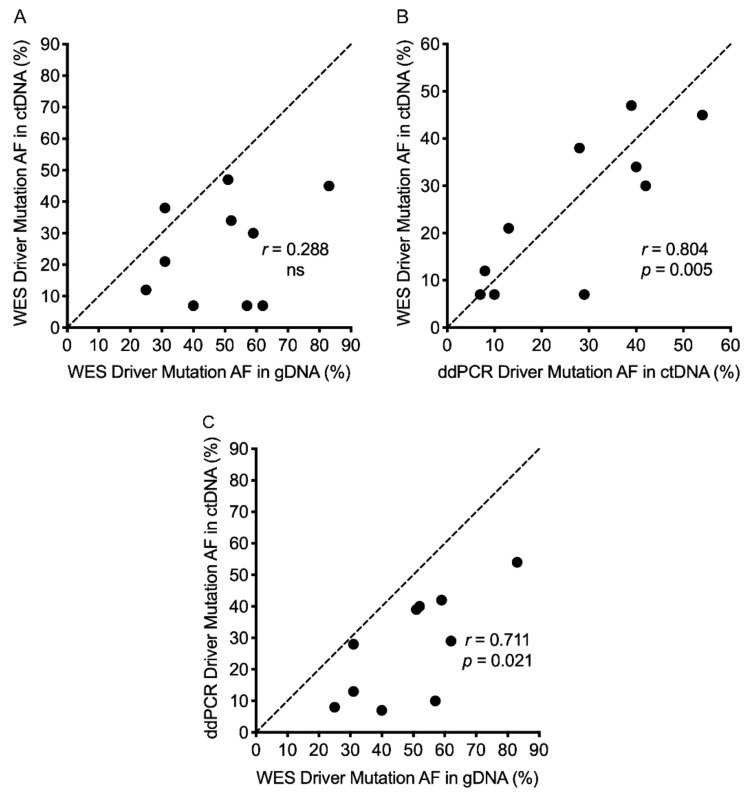
Degree of Pearson correlation based on the mutant allele frequency of the driver mutation in melanoma patients. (**A**) WES of genomic DNA (gDNA) versus WES of circulating tumor DNA (ctDNA). (**B**) ddPCR analysis of ctDNA versus WES of ctDNA. (**C**) WES of gDNA versus ddPCR analysis of ctDNA. Abbreviations: ns, not significant.

**Table 1 cancers-11-01905-t001:** Patient, treatment and biopsy characteristics.

Patient	Age (Years)	Sex	Tumor Biopsy Site	Current Treatment When Tissue Biopsy Procured	Current Treatment When Liquid Biopsy Procured	Time Between Biopsies (Months) *	Mutation	Stage (AJCC 8th ed [19])	Serum LDH **	OS (Months)
1	76	M	Brain	PROG pembro	PROG pembro	Concurrent	BRAF V600E NRAS Q22K	M1d	NA	4.8
2	61	F	Inguinal LN	PROG combi-DT	PROG combi-DT	6	BRAF V600E	M1c	<ULN	Alive (33.7)
3	58	M	Brain	PRE nivo	PROG nivo	2	BRAF V600E	M1d	>2× ULN	19.2
4	48	M	Thigh SC	PROG ipi + nivo	PROG ipi + nivo	1	BRAF V600E	M1d	<ULN	20.7
5	56	M	Inguinal LN	PRE combi-DT	PROG combi-DT	4	BRAF V600K	M1c	>2× ULN	10.4
6	70	M	Liver	PRE nivo	PRE nivo	Concurrent	BRAF G466E	M1d	>2× ULN	1.2
7	37	F	Ovary	PROG ipi + pembro	PROG ipi + pembro	4	NRAS Q61K	M1c	>1× ULN	25.7
8	65	M	Abdominal LN	PROG ipi + pembro	PROG ipi + pembro	Concurrent	NRAS Q61K	M1c	<ULN	Alive (52.2)
9	65	M	Liver	PROG nivo	PROG nivo	3	NRAS Q61R	M1c	>1× ULN	20.4
10	69	M	Brain	PRE pembro	PROG pembro	6	NRAS Q61R	M1c	<ULN	11.6

* In patients 2, 3, 4, 5 and 10, tumor biopsies were taken prior to the liquid biopsies and in patients 7 and 9, tumor biopsies were taken after the liquid biopsies. In patients 6 and 8, the liquid and tissue biopsies were taken at the same time. ** Taken at time of liquid biopsy. AJCC, American Joint Committee on Cancer; F, female; M, male; OS, overall survival; ULN, upper limit of normal; NA, not available; LN, lymph node; SC, subcutaneous; LN, lymph node; PRE, prior to systemic therapy; PROG, at time of progression on treatment; nivo, nivolumab (anti-PD1); pembro, pembrolizumab (anti-PD1); ipi, ipilimumab (anti-CTLA4); combi-DT, combination dabrafenib (BRAF inhibitor) and trametinib (MEK inhibitor).

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
