# Peer review of "Analysis of the Whole-Exome Sequencing of Tumor and Circulating Tumor DNA in Metastatic Melanoma"

_cancers, 2019, doi:10.3390/cancers11121905_

Round 1

Reviewer 1 Report

The work from Diefenbach et al. attempts to validate Whole Exome Sequencing of cfDNA by comparison to tumor gDNA in metastatic melanomas. The paper is of general technical interest, but some concerns are present as below.

Main points:

The authors need to clearly highlight the novelty of their work. I also suggest toning down “ctDNA and tumor genomic (g) DNA are rarely analysed together” as there are numerous reports in other cancer types and a few in melanoma. Can authors estimate the % of ctDNA that is in each of their cfDNA, and how this correlates with % tumor purity from the tumor gDNA? One metric that is missing is how evenly the genome is covered in cfDNA vs gDNA. A statistic such as coefficient of variance may be useful in this sense. The authors highlight a few interesting mutations in Table S4. In more than one patient some SNVs are found in much higher MAFs in cfDNA than in gDNA. How do the authors explain this observation? Following from the previous point, how do the authors know which mutations are biologically relevant (actually coming from tumor cells)? Additionally, it seems odd that only 2 samples have clear bona fide melanoma-associated mutations (NRAS Q22K and CDK4 R24C), is this just coincidence, or is it possible other mutations were missed?

Minor Points:

Additional detail on ddPCR is needed in the text, it should explicitly say that it is only for BRAF or NRAS, otherwise it is rather confusing on first readthrough. The title of the paper is misleading, as no landscape is presented. I suggest to instead highlight the technical value of the paper in the title (something like “Comparison of…”).

Reviewer 2 Report

The manuscript by Diefenbach Russell and colleagues deals with an important topic related to the analysis of circulating tumor DNA (ctDNA) as an important prognostic tool to potentially identify disease-associated mutations, tumor progression as well as treatment resistance.

The authors performed a comparative whole exome sequencing (WES) analysis on both genomic DNA and corresponding ctDNA. However, the manuscript has to be significantly improved. The number of considered patients should be increased to better define a correlation between the two DNA sources. Moreover, apart from the limited number of considered patients, it's difficult to draw conclusions since the tumor biopsy sites are so different for each patient and the corresponding blood samples were obtained  at different time points (some of them are concurrent to tumor biopsy, other were obtained after months). The authors should pay much more attention to the patient selection, otherwise the conclusions could be misleading.

As a minor revision, the authors should use the same acronym of circulating tumor DNA in the whole manuscript (in the Introduction section they indicate it as ctDNA, in other section as cfDNA).

Round 2

Reviewer 1 Report

Answers are fine

Reviewer 2 Report

The manuscript has been improved in the different sections (in particular Discussion section), and authors' response is quite convincing.

English grammar has been significantly improved, and  the manuscript is now acceptable.